# Proteomic Insights of Cowpea Response to Combined Biotic and Abiotic Stresses

**DOI:** 10.3390/plants12091900

**Published:** 2023-05-06

**Authors:** Daiane Gonzaga Ribeiro, Ana Carolina Mendes Bezerra, Ivonaldo Reis Santos, Priscila Grynberg, Wagner Fontes, Mariana de Souza Castro, Marcelo Valle de Sousa, Maria Eugênia Lisei-de-Sá, Maria Fatima Grossi-de-Sá, Octávio Luiz Franco, Angela Mehta

**Affiliations:** 1Centro de Análises Proteômicas e Bioquímicas Programa de Pós-Graduação em Ciências Genômicas e Biotecnologia, Universidade Católica de Brasília (UCB), Brasília CEP 71966-700, DF, Brazil; 2Embrapa Recursos Genéticos e Biotecnologia, PBI, Av. W/5 Norte Final, Brasília CEP 70770-917, DF, Brazil; 3Programa de Pós-Graduação em Ciências Biológicas (Biologia Molecular), Instituto de Ciências Biológicas, Campus Universitário Darcy Ribeiro-UnB, Universidade de Brasília, Brasília CEP 70910-900, DF, Brazil; 4Laboratório de Bioquímica e Química de Proteínas, Departamento de Biologia Celular, Universidade de Brasília, Brasília CEP 70910-900, DF, Brazil; 5National Institute of Science and Technology, INCT PlantStress Biotech, Embrapa, Brasilia CEP 70770-917, DF, Brazil; 6S-Inova Biotech, Universidade Católica Dom Bosco (UCDB), Campo Grande CEP 79117-900, MS, Brazil

**Keywords:** *Vigna unguiculata*, nematode, drought, resistance biomarkers

## Abstract

The co-occurrence of biotic and abiotic stresses in agricultural areas severely affects crop performance and productivity. Drought is one of the most adverse environmental stresses, and its association with root-knot nematodes further limits the development of several economically important crops, such as cowpea. Plant responses to combined stresses are complex and require novel adaptive mechanisms through the induction of specific biotic and abiotic signaling pathways. Therefore, the present work aimed to identify proteins involved in the resistance of cowpea to nematode and drought stresses individually and combined. We used the genotype CE 31, which is resistant to the root-knot nematode *Meloidogyne* spp. And tolerant to drought. Three biological replicates of roots and shoots were submitted to protein extraction, and the peptides were evaluated by LC-MS/MS. Shotgun proteomics revealed 2345 proteins, of which 1040 were differentially abundant. Proteins involved in essential biological processes, such as transcriptional regulation, cell signaling, oxidative processes, and photosynthesis, were identified. However, the main defense strategies in cowpea against cross-stress are focused on the regulation of hormonal signaling, the intense production of pathogenesis-related proteins, and the downregulation of photosynthetic activity. These are key processes that can culminate in the adaptation of cowpea challenged by multiple stresses. Furthermore, the candidate proteins identified in this study will strongly contribute to cowpea genetic improvement programs.

## 1. Introduction

In agricultural areas, cultivated plants can be simultaneously exposed to a wide range of biotic and abiotic stress, such as pathogen infection and drought. Climate change scenarios foresee increases in the earth’s temperature and a decrease in the rainfall regime, which can drastically affect the yield of important crops, including cowpea [1]. Cowpea (*Vigna unguiculata* L. Walp.) is a legume widely consumed all over the world. Vigna grains have high amounts of protein and vitamins, representing an important food source for the populations in these regions. Cowpea also plays an important role in soil fertilization due to the plant’s ability to fix atmospheric nitrogen in association with rhizobial bacteria [2]. However, nematode infestation and water deficits are among the main factors limiting cowpea productivity. The disease caused by root-knot nematodes (RKN) can impair water and nutrient absorption, causing leaf chlorosis and damaging the root system [3]. A water deficit restricts several aspects of vegetative growth, causing a series of biochemical, physiological, and morphological responses [4]. The effects of cross-stress, therefore, can severely affect cowpea development and productivity.

Plants have developed several mechanisms to cope with stress, and the study of the responses to single stresses is quite evolved. However, in the case of simultaneous biotic and abiotic stresses, plant responses are more complex, as they include the interaction of two living organisms combined with stress conditions [5]. It has been reported that the molecular responses of plants subjected to combined stresses may be completely different from their responses to individual stresses [6,7]. Studies have shown that plant responses to multiple stresses involve the activation of kinases that regulate specific genes, transcription factors, reactive oxygen species, heat shock proteins, and the crosstalk of several hormonal signaling pathways [8,9,10,11].

*V. unguiculata* is, in general, resistant to *M. incognita* and tolerant to a water deficit [12,13]. Cowpea resistance to root-knot nematodes seems to be controlled by the dominant gene Rk, which was proven effective against some RKN isolates but did not show any effectiveness against all RKN populations [14,15]. Regarding drought tolerance, studies on cowpea have associated several genes and multiple quantitative trait loci (QTL) with drought tolerance [16,17,18,19]. However, these studies represent individual assessments of cowpea-nematode interactions and cowpea response to drought. Although key mechanisms involved in RKN resistance and drought tolerance, such as activation of R genes and oxidative stress/ubiquitination, respectively, were reported [20,21,22,23], to the best of our knowledge, there are still few studies investigating plant responses to both stresses simultaneously.

Prospective studies using proteomic approaches can bring important contributions to understanding the molecular mechanisms involved in the defense response of plants to co-applied stresses. Furthermore, the proteomics approach is a promising tool for crop improvement programs as it allows for the global evaluation of total proteins, the final product of gene expression, from samples under different biological conditions. Therefore, the objective of this work was to identify proteins potentially involved in *V. unguiculata* resistance to *M. incognita* and drought tolerance, individually and combined (cross-stress). To evaluate the plants, we performed physiological (gas exchange) and shotgun proteomic analyses. Discovering the complex modulation of plant responses to simultaneous biotic and abiotic stresses may contribute to the development of cultivars resistant to multiple stresses, which often occur in the field.

## 2. Results and Discussion

### 2.1. Phenotypic Evaluation and Gas Exchange of Vigna unguiculata in Response to M. incognita and a Water Deficit

Plants subjected to a water deficit showed leaf wilting and retardation of vegetative growth during progressive water loss irrespective of nematode inoculation as compared to the control (Figure 1a,c,d). Inoculation with nematodes, on the other hand, caused leaf yellowing regardless of a water deficit (Figure 1a,b,d), as expected [24,25].

One of the main physiological strategies to cope with drought and prevent water loss in plants is stomatal control. The low water availability in the soil induces the closure of stomata through the induction of root signals, consequently causing a reduction in gas exchange and carbon fixation rates [26]. According to our results, we can observe a reduction in transpiration (*E*), stomatal conductance (*g*_s_), and net photosynthetic rates (*An*) in nematode and drought-exposed plants (ND) and in plants only under drought (D) from six days after treatment (DAT) (Figure 2B–D). In addition, plants submitted exclusively to D showed even lower rates of *An* when compared to the combined stress treatment, showing reductions of ~30% at 10 and 72% at 12 DAT. It was possible to observe an increase in intercellular CO_2_ concentrations (*Ci*) in response to water limitation in the last days of measurement (Figure 2A). Increases in *Ci* as a response to drought were previously reported in severely water-stressed cowpea plants [27]. As stated before, plant responses to a single stress cannot be extended to combined stress in most cases. In the literature, studies report biotic and abiotic stresses either enhancing or reducing plants’ tolerance when co-applied [28,29]. Plant responses will depend on many factors, such as plant species, stage of development, and intensity of the stressor factor [7]. Similar to our results, other authors also reported the reduction of *E*, *gs*, and *An* in plants subjected to single and combined stresses [30,31]. However, contrary to our results, the effects of the cross-stress (drought + inoculation with cowpea severe mosaic virus) in cowpea at an early stage of infection led to a progressive reduction of *C_i_*, whereas *An* was reduced by the stresses but remained similar when comparing the values of single and combined stresses [12].

In our study, nematode inoculation induced higher *An* rates when compared to control plants, presenting increases of ~1.4, 0.5, and 0.3-fold changes at 6, 8, and 10 DAT, respectively (Figure 2). It is known that *M. incognita* is an obligate parasite and, therefore, needs the cellular content of plants to feed. The presence of the parasite may stimulate the cellular machinery of the plant increasing energy production. On the other hand, control plants showed regular rates of the aforementioned physiological processes, producing energy for their development.

### 2.2. Protein Profile of Vigna unguiculata in Response to Combined Stress

Plants have developed a series of morphological, physiological, and molecular adaptations to survive in adverse conditions. The higher complexity of plants’ responses to combined biotic and abiotic stresses is due to the involvement of two living organisms, the plant and the pathogen, accompanied by the abiotic factor. Therefore, to better understand the mechanisms of resistance of cowpea to combined stress, the protein profiles of the roots and shoots were investigated. Proteomic analysis of cowpea roots and shoots revealed, respectively, 570 and 470 proteins differentially abundant in the three conditions evaluated (N, D, and ND). Principal component analysis and heatmaps were carried out for stress and control conditions, showing greater variability in the biological replicas under N (roots) and ND (shoots) (Figure 3). Gene ontology analysis showed several important biological processes in which differentially abundant proteins are involved, such as homeostatic processes, cell wall organization, transport, photosynthesis, and response to stimuli, among others (Figure 4 and Figure 5). In this study, important proteins from cowpea roots reported to be involved in common defense responses to biotic and abiotic stresses, such as the activation of the antioxidant mechanisms, production of NB-LRR proteins, and accumulation of secondary metabolites, were identified. Interestingly, some specific defense responses of cowpea subjected to cross-stress were also observed, such as the activation of the jasmonic acid hormonal signaling pathway and the intense production of PR proteins. These and other proteins appear to be important in activating efficient mechanisms of adaptation to cross-stress and will be discussed in more detail below.

#### 2.2.1. Defense Proteins Orchestrate Cowpea Adaptation Mechanisms to Cross-Stress

The NBS-LRR proteins (nucleotide-binding site leucine-rich repeat) are encoded by one of the largest families of genes involved in plant disease resistance and recognize pathogen-derived effector proteins [32,33]. In addition to NBS-LRR, pathogenesis-related proteins (PR) are also essential for plant defense against pathogens. PRs tend to accumulate in the infected tissue, protecting plants from further infections. In this study, increased LRR and PR proteins were identified in ND plants (Table 1 and Table 2), both in the shoots and roots (LRR receptor-like serine/threonine-protein kinase FLS2, Log2FC (FC): 12; chitinase, FC: 6; pathogenesis-related protein, FC: 5; thaumatin, FC: 3). Previous studies have identified NBS-LRR and PRs proteins involved in resistance to both biotic and abiotic stresses when evaluated individually [34,35,36], but little is known about the presence of these proteins under cross-stresses [10,37]. A recent study reported that the overexpression of an *A. stenosperma* endochitinase (*AsECHI*) induces resistance to RKN and a water deficit, isolated and combined. The higher drought tolerance of these plants was achieved concomitantly with a reduction of ~30% in infection by RKN [38]. In this context, the increased NB-LRR proteins modulated in ND may activate cascades of effective defenses against both stresses. ND plants also seem to increase the production of PRs to minimize the oxidative effects caused by a water deficit and nematode infection. Interestingly, we identified a chitinase protein only under ND. This protein can be a key candidate potentially involved in RKN resistance and drought tolerance. This protein was not identified in cowpea inoculated only with nematodes. The proteins mentioned above are included in the stimulus response category (Figure 4 and Figure 5).

Other proteins related to cowpea resistance were increased in the roots (Table 1) of N, D, and ND plants (CPRD86, CPRD14, and CPRD2) and in the shoots (Table 2) of ND plants (CPRD2 and dehydrin). Cowpea clones responsive to dehydration (CPRD) proteins have already been identified in other cowpea varieties, such as EPACE [39,40]. Dehydrin proteins play an important role in dehydration tolerance, as they help to increase water uptake and water holding capacity, favoring the maintenance of photosynthetic rates and consequently reducing the production of reactive oxygen species (ROS) [41,42]. Even though the responsiveness to dehydration proteins is primarily related to single drought resistance, our results showed its increase under both cross and single stresses, appearing to be a common cowpea resistance response. Overall, our results suggest that some resistance mechanisms of cowpea subjected to drought and cross-stress seem to be shared and that other mechanisms seem to be unique to ND, such as the intense activation of PRs proteins identified only in ND. The imposed water deficit may increase cowpea resistance to RKN. The proteins involved in dehydration are included in the protein binding category (Figure 4 and Figure 5).

#### 2.2.2. Antioxidant System Drives Plant Adaptation to Combined Stress

Reactive oxygen species are important signaling molecules of the signal transduction pathway that triggers stress defense responses, but the exacerbated production of these molecules can cause cellular damage [43]. Biotic and abiotic stresses cause dysregulation of the plant metabolism, which results in increased ROS generation affecting cellular redox homeostasis. ROS react with different cellular structures, such as the nucleus, proteins, and membranes, impairing their integrity, so its excessive production must be controlled by antioxidant systems to prevent damage and cell death. The activation of antioxidant enzymes crucial in the efficient elimination of ROS has become an essential marker of plant adaptation [44]. In this study, many antioxidant proteins were identified both in the shoots and in the roots of cowpea, especially under conditions of D and ND (Figure 4 and Figure 5—oxidoreductase activity category). Among the increased antioxidant proteins, we can highlight L-ascorbate peroxidase, glutathione transferase, and superoxide dismutase with fold-changes raging between two and nine (Table 1 and Table 2). Ascorbate peroxidase and glutathione transferase are antioxidant enzymes responsible for protecting the plasma membrane of cells [45,46]. In addition, it has been reported that one of the main antioxidant compounds strongly linked to plant adaptation to drought is ascorbate [47]. Other studies also reported the identification of several antioxidant proteins in plants subjected to both drought [48,49] and combined abiotic stresses [50,51] but there are few reports of antioxidant proteins in combined biotic and abiotic stresses [52,53]. Given the importance of antioxidant defense for stress resistance, it is notable that the activation of the ROS detoxification pathway in cowpea plants is key to preventing cellular damage caused by drought and root-knot nematodes.

#### 2.2.3. Defense Responses Mediated by Hormonal Signaling in Cowpea

Plants do not have an adaptive immune system, but they do have a signaling network to ensure growth and development as well as adaptation to adverse conditions. Growth regulators play a crucial role in the reprogramming of complex mechanisms of adaptation to stresses, such as abscisic acid (ABA), jasmonic acid (JA), salicylic acid (SA), and ethylene (ET) [37,54,55]. ABA and JA are strongly related to stress tolerance since ABA-mediated signaling allows the regulation of stomatal closure, and JA-mediated signaling triggers plant defense responses through its accumulation in leaves [56,57]. Although JA is known to be related to biotic stress, many other studies reinforce the role of this phytohormone in response to drought, showing a significant and rapid increase of JA in plants under water stress conditions [58,59]. Our study identified the increase of some proteins involved in signaling via JA (Lipoxygenases, FC: 2–3) and ABA pathways (protein containing Bet_v_1 domain, FC: 2–6 and annexin, FC: 1.8) in plants subjected to D and ND in both the roots and shoots (Table 1 and Table 2). Studies on the evaluation of cross and individual stresses in Arabidopsis and tomato reported that the hormonal signaling pathways and the expression of defense genes are different for each stress [5,60]. Another report involving the same treatments in peanuts showed that JA and ABA were activated in response to RKN and drought stress, respectively, while ET was activated in *Arachis* plants subjected to cross-stress (nematode and drought) [38]. Here, we can observe that through the crosstalk network, JA and ABA phytohormones may be working together to regulate cowpea tolerance to drought and combined stress.

Interestingly, our study identified several proteins related to the abscisic acid pathway that were highlighted in cowpea tolerance to drought and cross-stress. These data corroborate the physiological analysis of stomatal conductance (g_s_), which showed reduced rates on all evaluated days, with a significant reduction on the last day.

#### 2.2.4. Other Proteins’ Important in the Process of Adaptation to Drought and Multiple Stresses

During PTI and ETI induction, some signaling cascades are rapidly initiated, such as the activation of MAP kinases, production of reactive oxygen species, and transcriptional reprogramming, among others [61]. In plants, stimuli caused by adverse conditions are detected by specific receptors on the plasma membrane, initiating the signaling of the lipid cascade. Phospholipid-based signaling cascades are extremely important in signal transduction as phospholipids activate communication with the immune system preventing invasion by pathogens [62,63]. The main components of this signaling cascade are phospholipase C (PLC) and phospholipase D (PLD), which are also responsible for the turnover of phospholipids along with diacylglycerol kinase (DAK) [64]. Some studies have identified phospholipases involved in plant resistance against fungi and bacteria (mainly in effector recognition) [65,66], as well as in ABA-induced water stress tolerance [67,68,69]. In this study, some phospholipase C and phospholipase D (FC: 1.1-5, respectively) proteins were identified in cowpea plants subjected to D and ND in both roots and shoots (Table 1 and Table 2). Thus, it is likely that activation of signaling via phospholipids in cowpea plants is directly related to stomatal closure and reduced transpiration, as well as cell wall-based defenses against pathogenic infections. Protein phospholipases are included in the catalytic activity category (Figure 4 and Figure 5).

Secondary metabolites also play an important role in plant defense responses. Flavonoids are a large class of secondary metabolites present in plants and have several functions, such as plant development through the control of auxin, antioxidants, chemoattractants, and defense compounds, among others. Studies report that flavonoids can induce quiescence by decreasing the movement of nematodes, modifying their migration towards the root, repelling, and even killing them [70,71]. Flavonoids are also important in response to a water deficit, and their accumulation is often associated with drought resistance [72,73]. In our study, proteins involved in the flavonoid biosynthesis pathway were increased only in cowpea roots subjected to D (chalcone synthase, isoflavone reductase) and ND (flavonol 3-O-methyltransferase). Some authors associated the increase in plant flavonoids with abiotic stresses tolerance [74,75], but reports of alterations in metabolites involved in response to combined stresses are rare [76]. Cowpea plants subjected to D and ND seemed to accumulate secondary metabolites, such as flavonoids, important to eliminate cells infected by the parasite and damaged by drought. The proteins mentioned above were not identified in the cowpea-nematode interaction (N) in our study.

#### 2.2.5. Reduction of Photosynthetic Proteins in Response to Combined Stress in Cowpea

Photosynthesis is one of the primary processes altered by exposure to stress. Due to its direct relation with energy production, reductions in photosynthesis, depending on the severity, may lead to reduced growth, yield, and even plant death [77]. In the shoots of cowpea, D differentially induced the increase of proteins associated with leaf photochemistry (Table 2), such as chlorophyll a-b binding protein (FC: 4.47), involved with the capture and delivery of excitation energy to the photosystems and a protein identified as photosystem II stability/assembly factor (FC: 1.3), essential for photosystem II (PSII) biogenesis. It is known that PSII is highly susceptible to excessive excitation energy and elevated levels of ROS and the rates of its damage and repair of PSII control photoinhibition [78]. Photoinhibition is reversible in its early stages, and the repair of PSII is mainly associated with the *de novo* biosynthesis of PSII photosynthetic proteins [79]. In contrast, two other leaf photosynthesis-related proteins were decreased due to the exposure of plants to drought: the photosystem I (PSI) subunit PsaN (FC: 1.8), involved in the docking of the mobile electron carrier plastocyanin [80,81], and the subunit 1 of chlorophyll a-b binding protein (FC: 1.3), also involved in the light-harvesting complex [82].

The simultaneous application of the stresses (ND), however, induced a completely different response. Among the differentially abundant proteins in the shoots of *V. unguiculata* under combined stress, 172 were decreased (Appendix A), with the coordinated decrease of several major photosynthetic proteins involved in the light reactions and carbon assimilation with at least 2-fold change (Table 2), including several proteins involved in the biosynthesis of chlorophyll, such as: magnesium-protoporphyrin IX monomethyl ester (oxidative) cyclase (FC 1.8), magnesium protoporphyrin IX methyltransferase activity (FC: 2.72), NADPH-protochlorophyllide oxidoreductase (FC: 2.2), and the enzyme geranylgeranyl reductase (FC: 2.4); two proteins involved in energy capture and transfer chlorophyll a/b binding proteins in the light-harvesting complex I (FC: 3.3 and 3.2, respectively); PSII oxygen-evolving enhancer protein units 1 (FC: 1.2), 2 (FC: 3.3), and 3 (FC: 2.2); photosystem I chlorophyll a apoproteins A1, PSI-subunit VI, and the PSI iron-sulfur center were down-regulated with FC of 2.4, 2.0, and 1.5, respectively; two components of the electron carrier cytochrome b6-f complex, cytochrome f (petA; FC: 1.5) and cytochrome b6 (petB, FC: 1.8); the chloroplastic enzyme ferredoxin-NADP reductase (FC: 1.79); and the subunits alpha and beta of ATP synthase (FC: 1.1 and 0.9); among other. A fragment of the molecular chaperone of rubisco, rubisco activase, and ribulose bisphosphate carboxylase large chain were also reduced by FC 1.4 and 0.8, respectively. A progressive reduction of rubisco activity was reported under high temperatures (>35 °C) due to a decrease in rubisco activase activity [83]. The proteins involved in photosynthesis are represented in the cellular metabolic process category (Figure 5).

Thus, it was possible to identify that under single and cross-stress, cowpea plants adopted different strategies to cope. While under single stress, plants seemed to increase sink capacity and invest in the repair of PSII photodamaged units to try and maintain carbon fixation rates, while under combined stress, we observed a coordinated decrease of light and assimilatory reactions. The coordinated decrease of photosynthetic transcripts involved in leaf photochemistry, carbon reduction, and pigment synthesis was reported as a defense mechanism of plants against biotic stresses. The authors hypothesized that the turnover of photosynthetic proteins allows the redirection of plants’ energy and reallocation of nitrogen to the induction of defense proteins against the pathogen while maintaining the losses in carbon assimilation to only a moderate level [84]. In our study, the downregulation of photosynthetic proteins was observed exclusively under the ND condition (Table 2). Even though the reduced activity of photosynthesis lowered carbon fixation rates and subsequently negatively affected plant productivity, in terms of plant biology, it may also represent an important strategy to cope with the stresses, indicating that the damage caused by the production and accumulation of toxic products of photorespiration and/or reactive intermediates threatening to cause permanent photodamage could outweigh the benefit of sustaining the high carbon fixation rates [85]. Both hypotheses are consistent with our gas exchange results. On the one hand, we can observe a reduction in photosynthetic assimilatory rates (*An*) in all plants submitted to water restriction (D and ND) when compared to control plants from 10 and 12 DAT concomitant with changes in the protein profile, characterized by the increased expression of pathogen defense-related proteins under ND. On the other hand, levels of *An* were severely reduced at 12 DAT under D when compared to ND (showing reductions of 85%), indicating that in the long term, the coordinated reduction of photosynthetic proteins might be a more efficient strategy to cope with drought and avoid photodamage.

## 3. Material and Methods

### 3.1. Plant Material and Growing Conditions

The cultivar of *V. unguiculata* CE 31, which is resistant to *M. incognita* race 3 and tolerant to water deficits [12,13], was used in this study. These authors reported that cowpea plants were challenged with *M incognita* (race 3) and collected after sixty days of cultivation. The egg mass index analysis was scored according to [86], and the degree of resistance was determined according to [87]. Furthermore, cowpea CE 31 has been reported as tolerant to drought in other works and used in the evaluation of other combined stresses [12,22]. 

Seeds were disinfected with 3% (*v*/*v*) sodium hypochlorite for 5 min and soaked in distilled water for 20 min. Germination was carried out using Germitest^®^ paper (neutral pH, 28 × 38 cm), and the seeds were kept in the dark for the first three days until the seedlings’ emergence. Seedlings were transplanted into 500 mL plastic cups containing a mixture of autoclaved substrate and sand (1:1) and kept in a growth chamber under a photoperiod of 16 h/8 h (light/dark) and a temperature of 25 °C. During the first 20 days after transplantation, the plants were irrigated with tap water, and from the 21st day on, plants were watered only with Hoagland’s nutrient solution (1:10) [88]. We evaluated plants inoculated with nematodes (N), plants submitted to a water deficit (D), plants inoculated with nematodes and a water deficit simultaneously (ND), and control plants (C). Control plants were not inoculated with nematodes and received a water regime to maintain 70% of FC (field capacity). Treatments were initiated 25 days after transplantation, and three biological replicas were used for each treatment. In the first three days after nematode inoculation and interruption of irrigation, no physiological measurements were performed, as this time was estimated for parasite penetration into plant roots. Gas exchange measurements were performed during the last nine days of the experiment, and all treatments were evaluated and compared to the control. Whole plants were collected for proteomic analysis. Plants were separated into shoots (all leaves and stems) and roots (whole root) and collected 12 days after treatment when they presented an average of 2 to 3 trifoliate leaves.

### 3.2. Plant-Nematode Interaction Assay (N)

The *Meloidogyne incognita* race 3 population was previously cultivated in tomato roots (cv. Santa Cruz) in a greenhouse. To extract the nematodes, the roots were triturated in a blender with 0.5% (*v*/*v*) sodium hypochlorite, as previously described [89]. Second-stage juvenile nematodes (J2) were collected using modified Baermann funnels, and nematode counting was performed with a light microscope using Peter’s counting slides. Cowpea plants were inoculated with 1000 J2 25 days after transplantation to evaluate the plant-pathogen interaction. Shoots and roots of plants inoculated with nematodes were collected 12 days after inoculation and stored at −80 °C for further evaluation.

### 3.3. Water Deficit Assay (D)

Field capacity (FC) was determined considering the weight difference between the wet soil after saturation and free drainage and the dry soil. FC maintenance was measured daily in all plants by weighing the cups and replacing the volume of water lost through transpiration using a scale with a capacity of 20 kg. Twenty-five days after transplantation, irrigation was interrupted for 12 consecutive days. During this period, soil water gradually reduced to 25% of FC. Shoots and roots of the plants subjected to drought were collected 12 days after irrigation interruption and stored at −80 °C for further evaluation.

### 3.4. Cross-Stress Assay (Nematode and Drought; ND)

Cowpea plants at 25 days of age were inoculated with 1000 J2 and subjected to drought simultaneously. Irrigation interruption was performed for 12 consecutive days until the plants reached 25% of FC. The shoots and roots of the plants were collected 12 days after nematode inoculation and irrigation interruption and stored at −80 °C for further evaluation.

### 3.5. Evaluation of Gas Exchange

Gas exchange measurements were performed in the second fully expanded trefoil of *V. unguiculata* in the mornings (from 8 am to 12 pm.) using a portable infrared gas analyzer system (IRGA LCpro-SD—ADC BioScientific, Hoddesdon, England). Intercellular CO_2_ partial pressure (*Ci*), transpiration rate (*E*), stomatal conductance (*gs*), and net photosynthesis rate (*An*) were recorded. The environmental conditions inside the IRGA chamber during the evaluation were: photosynthetic photon flux density (PPFD), 1000 µmol m^−2^ s^−1^; temperature, 28 °C; and ambient CO_2_ concentration. Gas exchange measurements were performed from the sixth day after nematode inoculation, a period sufficient for the penetration of the parasite into the roots of the plants. We presented data on gas exchange parameters at 6, 8, 10, and 12 days after treatment (DAT).

### 3.6. Statistical Analysis

The homogeneity of variances was previously analyzed using the Cochran homogeneity test. Whenever the data distribution was not homogeneous, box-cox transformation was used. After that, data were submitted to one-way analysis of variance (ANOVA), and in the presence of significant differences, means were compared using Tukey’s test with a 5% confidence level (*p* ≤ 0.05).

### 3.7. Protein Extraction, Digestion, and Desalting

Protein extraction was performed following Ribeiro et al., 2022 [21]. For the extraction of total proteins, 300 mg of plant tissue (root and shoots) were solubilized in an extraction buffer, and later phenol was added. The samples were agitated for 15 min and centrifuged at 10,000× *g* for 3 min. Proteins were precipitated in 0.1 M ammonium acetate in methanol and washed with 80% acetone. For protein digestion, the extracted proteins were solubilized with 60 µL of 50 mM ammonium bicarbonate (NH_4_HCO_3_ pH 8.5) and 25 µL of RapiGestTM SF—Waters (0.2% *v*/*v*). The samples were heated at 80 °C for 15 min, under agitation, and 2.5 µL of 100 mM dithiothreitol was added. For protein alkylation, 2.5 µL of 300 mM iodoacetamide was added, and the samples were kept at room temperature for 30 min in the dark. After reduction and alkylation, the proteins (approximately 80 µg) were hydrolyzed with trypsin (1 µg) and incubated in an oven at 37 °C for 19 h. Later, 10 µL of 5% trifluoroacetic acid (TFA) was added, and the samples were incubated again at 37 °C for 90 min. Samples were centrifuged at 10 °C for 30 min, and the supernatant was recovered and dried. Protein quantification was performed with Qubit^®^ Fluorometer (Invitrogen), according to the manufacturer’s manual. Subsequently, the samples were desalted following the protocol of Rappsilber et al., 2007 [90] with modifications [91]. First, the tips were prepared (P200 µL) with Empore^®^ C18 disks, with one disk in each tip. Second, the tips containing the disks were washed with 20 µL of methanol and centrifuged at 2000× *g* for 30 s. Third, 20 µL of POROS^®^ R2 resin in solvent B (0.1% formic acid/98% acetonitrile) was added, and the tips were centrifuged at 2000× *g* for 30 s. Each tip was washed two times with 20 µL of solvent A (0.1% formic acid/2% acetonitrile) followed by centrifugation. After tip preparation, the samples were solubilized in solvent A, inserted in the tips, and centrifuged at 2000× *g* for 2 min. The peptides were eluted with 20 µL of solvent B and dried in a vacuum centrifuge for 45 min. Then, the peptides were solubilized with 0.1% formic acid and injected into the ESI LC-MS/MS.

### 3.8. Chromatography and Mass Spectrometry Analysis

The peptides obtained were injected into a chromatographic system (Dionex Ultimate 3000 RSLCnano UPLC, Thermo, Waltham, MA, USA) configured with a trap column (3 cm × 100 µm) containing C18 particles with 5 µm, 120 Å (ReprosilPur, Dr. Maich GmbH, Ammerbuch, Germany). The samples were injected to obtain 2 µg in the column and submitted to a linear gradient of elution between solvents A (0.1% formic acid in 2% acetonitrile/water) and B (0.1% formic acid in 80% acetonitrile/water) from 2% B to 35% B for 155 min. The fractions separated in the chromatographic system were eluted directly at the ionization source of an Orbitrap Elite mass spectrometer (Thermo, USA) configured for data-dependent acquisition mode. MS1 spectra were acquired on the orbitrap analyzer with a resolution of 120,000 and a range between 300 and 1650 *m*/*z*. The 15 most intense ions were fragmented, generating MS2 spectra with a resolution of 15,000 [92,93]. The reanalysis of already fragmented ions was inhibited by dynamic exclusion, favoring the identification of less abundant peptides.

### 3.9. Quantitative and Qualitative Analysis of Proteins

Chromatogram alignment and peptide quantification were performed using the Progenesis QI for Proteomics software. Protein identification was performed using Peaks^®^ 7.0 software (Bioinformatics Solutions Inc., Waterloo, ON, Canada) with the following parameters: database obtained from the UniProt repository (Universal protein), filtered for *Vigna unguiculata* (Taxon ID 3917), and submitted to the removal of redundant sequences using the FASTAtools software (http://lbqp.unb.br/LBQPtools/; accessed on 15 July 2021). The search was performed based on de novo and PSM sequencing with the following parameters: tolerance for precursor mass of 10 ppm, fragments of 0.05 Da, tolerance of up to 2 missed cleavages, carbamidomethylation of cysteines as fixed modification, and oxidation of methionine as a variable modification. Protein identification was considered significant at a false discovery rate of less than 1% (FDR < 1%). Differentially regulated proteins were detected using the ANOVA test (*p* < 0.05) and fold change ≥1.1. Abundant proteins were submitted to multivariate statistical analysis—PCA and grouped according to relative abundance profiles. The gene ontology was obtained using pfam2go software (https://rdrr.io/github/missuse/ragp/man/pfam2go.html, accessed on 10 June 2022). Proteomic data is available via ProteomeXchange (PXD031824 and PXD031808).

## 4. Conclusions

We performed gas exchange and proteomic analysis of cowpea plants inoculated with nematodes and submitted to a water deficit, individually and simultaneously. According to the photosynthesis analysis, plants submitted to a water deficit and combined stress showed reductions in several gas exchange parameters such as *E*, *gs*, and *An* when compared to irrigated plants. Meanwhile, plants only inoculated with nematode presented higher values of *An* compared to control plants until 10 DAT. We also investigated the proteins and genes expressed in resistant cowpea (CE31) during *M. incognita* infection and a water deficit, and several differentially abundant candidates were identified. To the best of our knowledge, this is the first study that evaluates the protein profile of resistant cowpea subjected to biotic and abiotic stresses simultaneously, aiming to identify the main processes involved in plant resistance and elucidate the main defense strategies against the nematode and drought tolerance.

In the proteomic analysis of cowpea subjected to isolated and combined stresses, it was possible to observe shared mechanisms of resistance between the stresses, such as the activation of NB-LRR proteins and antioxidant activity. We also identified specific strategies induced by the cross-stress, such as the activation of the signaling mediated by jasmonic acid, the intense production of PR proteins, and a coordinated down-regulation of photosynthetic proteins. The amount and abundance of PRs proteins identified in cowpea subjected to cross-stress may indicate that the imposition of a water deficit can increase the resistance of cowpea to RKN. The regulation of biotic and abiotic responses in plants is complex; however, the differentially abundant proteins here identified (shared and/or specific) as a response to the stresses can help draw a clearer picture of the commonalities and differences in cowpea responses to single and cross stresses. Furthermore, our results suggest that the abiotic stress (drought) had a predominant effect over the biotic stress (nematode). It is noteworthy that the target proteins identified in this study could contribute to the genetic improvement of cowpea in order to develop cultivars resistant to multiple stresses. In addition, farmers could have a lower cost in grain production due to the reductions in irrigation demands and the use of pesticides.

## Figures and Tables

**Figure 1 plants-12-01900-f001:**
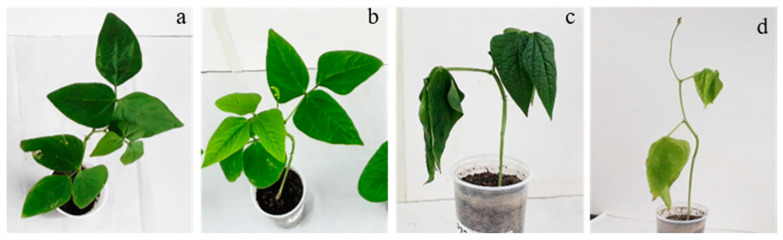
Morphological aspects of cowpea plants at 12 days after inoculation with nematodes (1000 J2) and interruption of irrigation (25% CF). Control CE 31 plants (**a**), inoculated with nematodes (**b**), submitted to water deficit (**c**), and inoculated with nematode and submitted to drought (**d**).

**Figure 2 plants-12-01900-f002:**
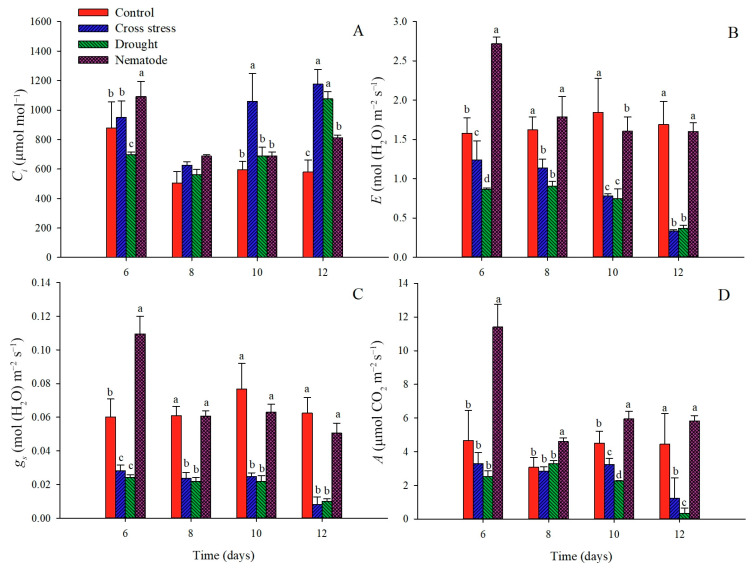
Intercellular CO_2_ concentrations, *Ci* (**A**), carbon transpiration rate, *E* (**B**), stomatal conductance, *gs* (**C**), and net photosynthesis rate, *An* (**D**) in leaves of *V. unguiculata* CE 31 inoculated with *M. incognita* (1000 J2) and/or submitted to water deficit (25% of field capacity) evaluated at 6, 8, 10, and 12 DAT. Each bar represents the mean of three biological replicates with standard deviation. Small caption letters represent comparisons among treatments using Tukey’s test with a 5% confidence level (*p* ≤ 0.05).

**Figure 3 plants-12-01900-f003:**
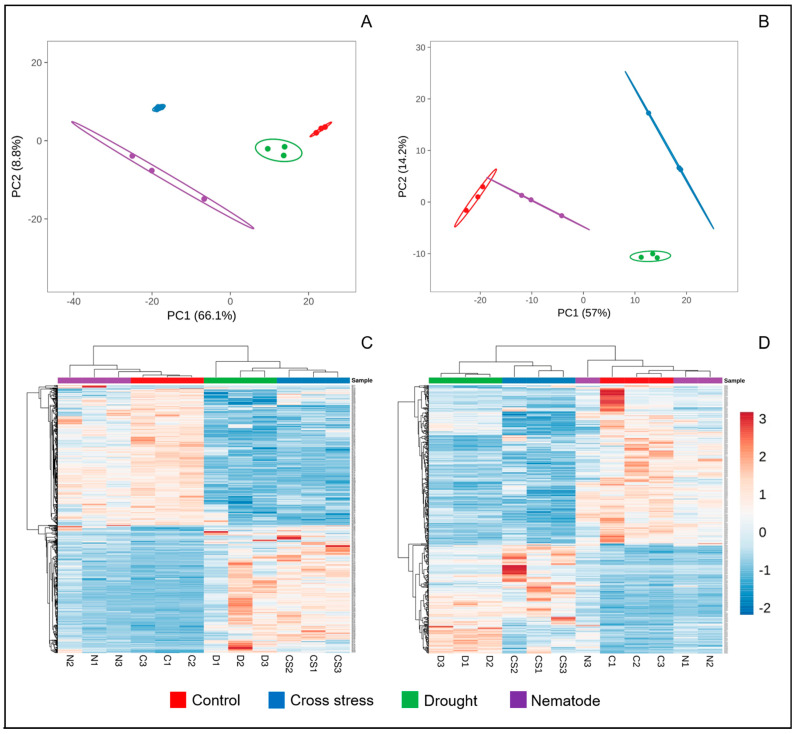
Principal component analysis (PCA) and heat maps of differentially abundant proteins in roots (**A** and **C**, respectively) and shoots (**B** and **D**, respectively) of cowpea.

**Figure 4 plants-12-01900-f004:**
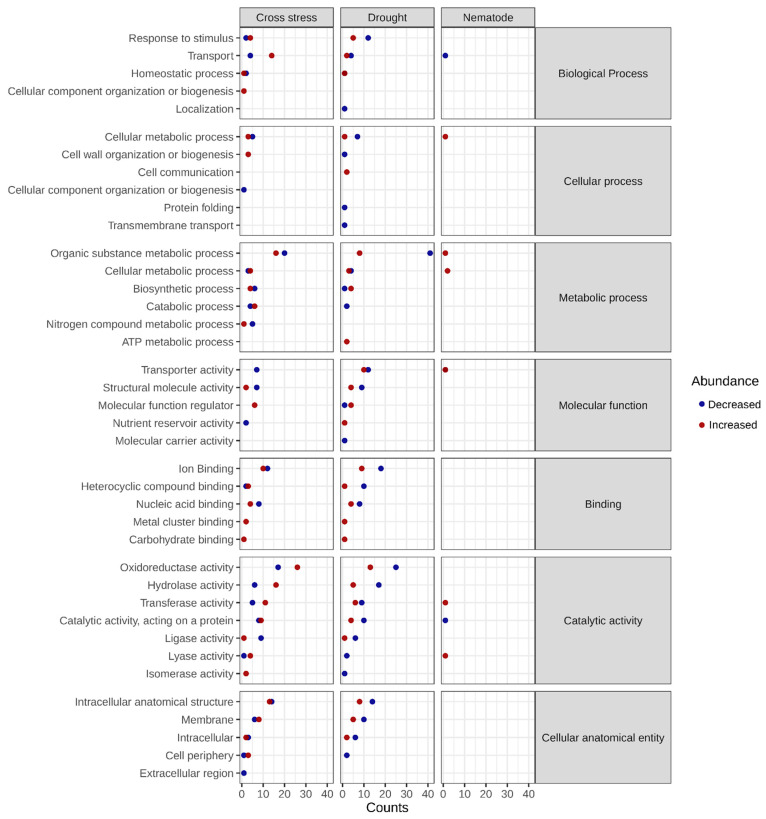
Bubble plot of the biological processes, molecular functions, and cellular components in which differentially abundant proteins of roots are involved. Increased proteins are represented by red bubbles and decreased by blue bubbles.

**Figure 5 plants-12-01900-f005:**
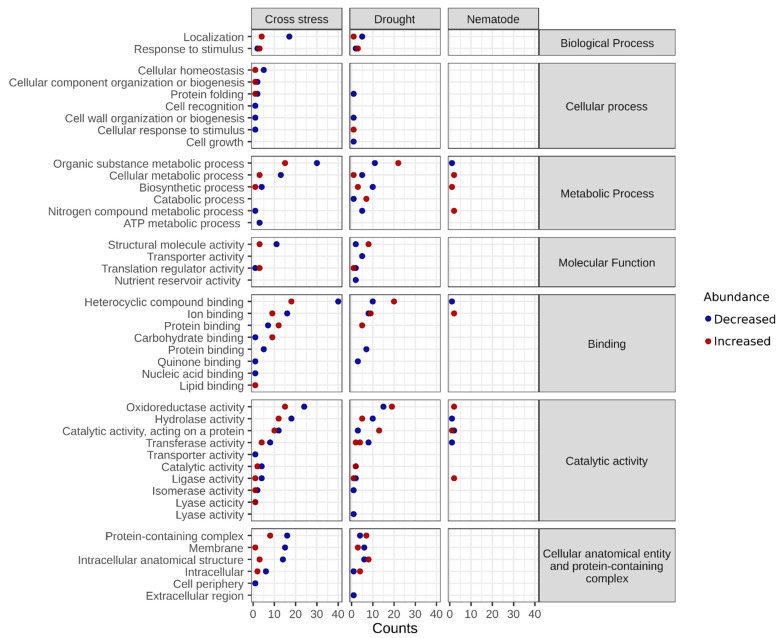
Bubble plot of biological processes, molecular functions, and cellular components in which differentially abundant proteins from shoots are involved. Increased proteins are represented by red bubbles and decreased proteins by blue bubbles.

**Table 1 plants-12-01900-t001:** Differentially abundant proteins of cowpea roots inoculated with nematodes and submitted to drought, individually and simultaneously, when compared to the control.

Biological Process	Protein ID (UniProt)	Description	Score	Anova (*p* ≤ 0.05)	Log2FC	Cross/Control	Drought/Control	Nematode/Control
Defense response	A0A4D6LDE6	LRR receptor-like serine/threonine-protein kinase FLS2	188.44	0.001	-	Exclusive		
	A0A4D6LDX5	LRR receptor-like serine/threonine-protein kinase FLS2	36.16	0.023	-	Exclusive		
	A0A4D6M0M0	Chitinase	68.29	0.008	6.29	Increased		
	A0A4D6KNF4	Pathogenesis-related protein 1	350.07	9.42 × 10^−5^	5.58	Increased		
Dehydration response	Q9AYM8	CPRD2 protein	119.93	0.001	3.22	Increased		
	P93700	CPRD14 protein	1000.73	0.002	1.7		Increased	
	Q9FS23	CPRD86 protein (Fragment)	121.7	0.025	6.11			Increased
Oxidative stress	A0A4D6LCC8	Glutathione S-transferase	50.39	0.017	9.43	Increased		
	Q9M7R2	Superoxide dismutase	510.36	0.001	3.74		Increased	
	A0A4D6MLJ0	Glutathione peroxidase	221.88	0.003	2.15		Increased	
Jasmonic acid biosynthetic process	A0A4D6MNE8	Lipoxygenase	132.89	1.36 × 10^−6^	3.72	Increased		
	A0A4D6MNU1	Lipoxygenase	84.89	0.003	3.53	Increased		
Response to abscisic acid	A0A4D6LN47	Bet_v_1 domain-containing protein	284.53	0.019	3.61	Increased		
	A0A4D6LBT9	Bet_v_1 domain-containing protein	29.69	0.019	6.73		Increased	
Lipid metabolic process	A0A4D6N7S0	Phospholipase D	1345.21	0.031	1.11	Increased		
	O04865	Phospholipase D alpha 1	424.29	0.013	1.06	Increased		
	A0A4D6MIW6	Phosphoinositide phospholipase C	51.84	0.009	5.77		Increased	
Flavonoid biosynthetic process	A0A4D6NPD2	Chalcone synthase	90.18	0.000	2.31		Increased	
	A0A4D6NRT4	Isoflavone reductase	840.96	0.006	0.77		Increased	
	A0A4D6NPC3	Flavonol 3-O-methyltransferase	407.45	0.002	3.86	Increased		

**Table 2 plants-12-01900-t002:** Differentially abundant proteins of cowpea shoots inoculated with nematodes and submitted to drought, individually and simultaneously, when compared to the control.

Biological Process	Protein ID (UniProt)	Description	Score	Anova (*p* ≤ 0.05)	Log2FC	Cross/Control	Drought/Control	Nematode/Control
Defense response	A0A4D6M0I2	LRR receptor-like serine/threonine-protein kinase FLS2	26.99	7.07 × 10^−5^	12.52	Increased		
	A0A4D6LBY3	Chitinase	247.88	0.005	2.14	Increased		
	A0A4D6LT51	Thaumatin	120.72	7.80 × 10^−4^	3.62	Increased		
Dehydration response	P93700	CPRD14 protein	566.23	1.98 × 10^−4^	1.64	Increased		
	A0A4D6NTQ2	Dehydrin	327.31	0.03	1.95	Increased		
Oxidative stress	A0A4D6MLG9	Glutathione transferase	320.25	2.174 × 10^−5^	2.94	Increased		
	A0A4D6LKZ9	Glutathione S-transferase	41.73	0.001	2.25		Increased	
	Q41712	L-ascorbate peroxidase	623.75	0.02	1.24	Increased		
	Q5QIA9	L-ascorbate peroxidase	221.68	0.001	2.01		Increased	
	A0A4D6N707	Superoxide dismutase	329.21	0.041	1.20			Increased
Jasmonic acid biosynthetic process	A0A4D6MNU1	Lipoxygenase	86.99	0.039	2.72	Increased		
	A0A4D6MNL5	Lipoxygenase	406.25	1.88 × 10^−4^	3.81		Increased	
Response to abscisic acid	A0A4D6LBT9	Bet_v_1 domain-containing protein	29.09	0.049	2.55		Increased	
	A0A4D6LB15	Annexin	1459.33	1.87 × 10^−5^	1.83	Increased		
Lipid metabolic process	A0A4D6N7S0	Phospholipase D	1182.19	5.98 × 10^−6^	1.68	Increased		
	A0A4D6MCA1	Phospholipase A1	67.94	0.017	2.87		Increased	
Photosynthesis	A0A4D6L6M3	Chlorophyll a-b binding protein, chloroplastic	288.64	0.008	1.32		Increased	
	A0A4D6LJ91	Photosystem II stability/assembly factor	730.69	0.014	1.86		Increased	
	A0A4D6L8F7	Photosystem I subunit PsaN	82.18	0.02	1.80		Decreased	
	A0A4D6M6X8	Magnesium-protoporphyrin IX monomethyl ester (oxidative) cyclase	50.37	1.49 × 10^−2^	2.72	Decreased		
	A0A4D6L976	Magnesium-protoporphyrin O-methyltransferase	252.45	5.47 × 10^−4^	2.26	Decreased		
	A0A4D6N7U7	NADPH-protochlorophyllide oxidoreductase	281.88	1.81 × 10^−5^	2.45	Decreased		
	A0A4D6KW86	Geranylgeranyl reductase	58.53	4.78 × 10^−4^	3.34	Decreased		
	A0A4D6MNX0	Chlorophyll a-b binding protein, chloroplastic	703.79	1.40 × 10^−4^	3.21	Decreased		
	A0A4D6NHQ4	Chlorophyll a-b binding protein, chloroplastic	95.34	0.04	3.32	Decreased		
	A0A4D6N658	Photosystem II oxygen-evolving enhancer protein 2	118.02	4.66 × 10^−3^	2.26	Decreased		
	A0A4D6NAD0	Photosystem II oxygen-evolving enhancer protein 3	232.75	0.03	1.27	Decreased		
	A0A4D6L0K2	Photosystem II oxygen-evolving enhancer protein 1	54.41	0.05	2.47	Decreased		
	A0A4D6M3U4	Photosystem I P700 chlorophyll a apoprotein A1	1060.97	1.40 × 10^−2^	1.51	Decreased		
	I2E2T7	Photosystem I iron-sulfur center	322.98	5.73 × 10^−4^	1.87	Decreased		
	J7EX90	Cytochrome b6	806.96	3.33 × 10^−3^	1.54	Decreased		
	I2E2Q5	Cytochrome f	341.3	5.15 × 10^−5^	1.79	Decreased		
	A0A4D6LQE0	Ferredoxin--NADP reductase, chloroplastic	2375.03	2.84 × 10^−5^	1.13	Decreased		
	I2E2P9	ATP synthase subunit alpha	3858.04	2.73 × 10^−3^	0.97	Decreased		
	I2E2M7	ATP synthase subunit beta	388.27	2.78 × 10^−6^	2.04	Decreased		
	A0A4D6MKZ9	Photosystem I subunit VI	637.47	5.69 × 10^−3^	1.42	Decreased		
	A6H596	Putative rubisco activase (Fragment)	38.23	5.76 × 10^−5^	4.47	Decreased		

## Data Availability

Proteomic data is available via ProteomeXchange (PXD031824 and PXD031808).

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
