# Peer review of "Proteomic Insights of Cowpea Response to Combined Biotic and Abiotic Stresses"

_plants, 2023, doi:10.3390/plants12091900_

Round 1
Reviewer 1 Report
A lot of research has been published concerning individual abiotic or biotic stresses, however, only few reports have dealt with combined biotic and abiotic stress which usually co-occur in field conditions and the outcome of such a combination could be either positive or negative, depending on timing, stress severity, plant and pathogen characteristics. Besides, the individual stresses cannot be extrapolated to characterize the stress combination which usually presents unique features. In this work the authors have chosen to analyze at the protein level the response of a relatively tolerant cowpea genotype to individual drought and root-knot nematode stresses as well as their combination. The results are interesting and novel. I have some remarks to be addressed.
The authors claim that the vigna genotype used in their study is tolerant to drought and root knot nematode infection, but without presenting evidence or previous research proving this tolerance. Fig 1 – clear stress effects are presented which are not convincing for stress tolernce.The focus of the research is to compare single stresses and cross stress but I wonder if proteins involved in stress response are really contributing to stress tolerance – how one can differentiate the impact?
The description of the methods needs some details. Line 103 – “during the last 9 days of experiment” – unclear. From the results we can see that photosynthetic parameters were followed in dynamics and for proteomics the samples were collected after 12-days treatments, when morphological changes became visible. Fig 1 - at the time of sampling the cowpea plants had 2-3 trifoliate leaves. Line 149 – which parts of the root and shoot were used for proteomic analyses? Different plant parts can have different proteomic signature, for ex. Young, fully developed and senescing leaves.
References should be carefully checked.
Refs – 76 and 76 – the same thing
Line 235 – refs [31,32]; line 239 – [17] – names and year?
Ref 50. Morales et al - Plants 2020, 9(1), 88; https://doi.org/10.3390/plants9010088
Ref 44 Laxa et al - Antioxidants 2019, 8(4), 94; https://doi.org/10.3390/antiox8040094
ref 32. Hosseini, ref 53 Muthusamy, ref 82- Vieira – not found in the text
Author Response
Manuscript. ID: plants-2326773
Title: Proteomic insights of cowpea response to combined biotic and abiotic stresses
Plants Journal
Dear Editor,
I would like to thank the reviewers and editor for all relevant suggestions and comments that have substantially contributed to the manuscript's improvement. All the changes suggested have been accepted or justified and are listed below. If you need any additional information about the manuscript, we are available to provide it.
Yours sincerely,
Angela Mehta
Comments from the editors and reviewers:
Reviewer #1:
A lot of research has been published concerning individual abiotic or biotic stresses, however, only few reports have dealt with combined biotic and abiotic stress which usually co-occur in field conditions and the outcome of such a combination could be either positive or negative, depending on timing, stress severity, plant and pathogen characteristics. Besides, the individual stresses cannot be extrapolated to characterize the stress combination which usually presents unique features. In this work the authors have chosen to analyze at the protein level the response of a relatively tolerant cowpea genotype to individual drought and root-knot nematode stresses as well as their combination. The results are interesting and novel. I have some remarks to be addressed.
-The authors claim that the vigna genotype used in their study is tolerant to drought and root knot nematode infection, but without presenting evidence or previous research proving this tolerance. Fig 1 – clear stress effects are presented which are not convincing for stress tolernce. The focus of the research is to compare single stresses and cross stress but I wonder if proteins involved in stress response are really contributing to stress tolerance – how one can differentiate the impact?
Reply: We included two references in the text that previously reported the genotype CE31 as tolerant to both stresses. More information about the tolerance of cowpea to the evaluated stresses was added in the text, line 90. Silva et al., 2016 using the same reported drought-tolerant cowpea variety also observed stress symptoms such as mild leaf wilting, due to the level of stress imposed (25% of pot capacity). Indeed it is not easy to differentiate the impact, however, we can infer the impact when comparing the protein profile of individual stresses with comobined stresses.
Regarding the importance of proteomic analysis in this scenario, under field conditions cultivated plants are often affected by biotic and abiotic stresses simultaneously. The study of individual stresses, both nematode resistance and drought tolerance, have already been described in other works, for several economically important plants, however, the molecular mechanisms of adaptation of these plants to multiple stresses are still not fully understood. Thus, the information acquired at the protein level can be valuable to decipher the molecular and physiological mechanisms involved in the adaptation of plants to multiple stresses.
-The description of the methods needs some details. Line 103 – “during the last 9 days of experiment” – unclear. From the results we can see that photosynthetic parameters were followed in dynamics and for proteomics the samples were collected after 12-days treatments, when morphological changes became visible. Fig 1 - at the time of sampling the cowpea plants had 2-3 trifoliate leaves. Line 149 – which parts of the root and shoot were used for proteomic analyses? Different plant parts can have different proteomic signature, for ex. Young, fully developed and senescing leaves
Reply: We report that the last 9 days of the experiment were evaluated because the first three days after inoculation we did not perform the physiological analysis. We estimate that a period of three days is sufficient for nematodes to penetrate plant roots. This information has been added more clearly in the text, line 108. Additionally, more detailed data, to allow the study reproducibility, was inserted in the material and methods section. Information about the vegetative stage of plants in the moment of harvest and plant parts collected has also been added to the text, line 114, new version.
-References should be carefully checked.
Refs – 76 and 76 – the same thing
Line 235 – refs [31,32]; line 239 – [17] – names and year?
Ref 50. Morales et al - Plants 2020, 9(1), 88; https://doi.org/10.3390/plants9010088
Ref 44 Laxa et al - Antioxidants 2019, 8(4), 94; https://doi.org/10.3390/antiox8040094
ref 32. Hosseini, ref 53 Muthusamy, ref 82- Vieira – not found in the text
Reply: All references have been checked and duly placed in the text.
Reviewer 2 Report
The manuscript by Ribeiro et al., provides features about the biotic and abiotic stress tolerance of Cowpea through proteomic analysis. The manuscript is well written. Below are few comments/suggestions to the authors.
1. What is the level of biotic and abiotic stress tolerance in this plant? Did the authors consider this factor while eliciting stress?
2. If cowpea plants have tolerance against water deficit and RKN stress, why do they have shown leaf wilting and change in color?
3. In the heat map (Figure 3C) of roots, the expression pattern of D1 seems to be very different than D2 and D3, but the PCA plot does not show much variation. Authors should explain this. Is D1 an outlier?
4. Many proteins were found to be downregulated with both single and combined stress. This reviewer thinks that Authors did not focus much on the downregulated proteins.
5. ‘Differentially abundant’ seems to be misleading. Instead, ‘differentially regulated’ could be used. Also, the proteins that are increased in expression when compared to control can be mentioned as upregulated and proteins that are decreased in expression can be mentioned as downregulated.
6. What cutoff conditions were set to choose the differentially regulated proteins? For example, p value, fold change etc. Please mention it in the manuscript.
7. It is surprising that protein in response to stimulus is unaltered in nematode group. Is there a possible explanation for this?
8. Authors could use cytoscape analysis for visualizing the PPI network and network analysis of differentially expressed genes.
9. Authors can explain in detail how the knowledge gained from this study can be used in dealing with the biotic and abiotic stress confronted by other plants.
Author Response
Manuscript. ID: plants-2326773
Title: Proteomic insights of cowpea response to combined biotic and abiotic stresses
Plants Journal
Dear Editor,
I would like to thank the reviewers and editor for all relevant suggestions and comments that have substantially contributed to the manuscript's improvement. All the changes suggested have been accepted or justified and are listed below. If you need any additional information about the manuscript, we are available to provide it.
Yours sincerely,
Angela Mehta
Comments from the editors and reviewers:
Reviewer #2:
The manuscript by Ribeiro et al., provides features about the biotic and abiotic stress tolerance of Cowpea through proteomic analysis. The manuscript is well written. Below are few comments/suggestions to the authors.
- What is the level of biotic and abiotic stress tolerance in this plant? Did the authors consider this factor while eliciting stress?
Reply: The level of tolerance to biotic stress (nematode) was evaluated by Oliveira et al., 2012 and characterized as highly resistant. The study reports that cowpea plants were inoculated with second-stage juveniles (J2) of M. incognita (race 3) and collected sixty days after planting. Egg mass index was scored according to Taylor and Sasser et al. 1978 and the degree of resistance was designated according to Sasser et al., 1984. As for drought tolerance, other articles have already reported that CE 31 is drought tolerant and have already used this genotype in other combined stress experiments. The field capacity of 25% is considered moderate stress in cowpea (Carvalho et al., 2019), and was reported to cause significant reductions in photosynthetic rates as net photosynthesis and stomatal conductance and biomass accumulation in the cultivar CE-31 (Silva et al., 2016).
Additionally, previous experiments were carried out to determine the number of nematodes, the most suitable time for nematode inoculation and the order in which the stresses should be applied.
The information about the tolerance of cowpea to the evaluated stresses was added in the text, line 91.
- If cowpea plants have tolerance against water deficit and RKN stress, why do they have shown leaf wilting and change in color?
Reply: The cultivar used in the study was previously reported in other studies as tolerant to both stresses. We included two references in the text. Silva et al. (2016) using the same reported drought-tolerant cowpea variety also observed stress symptoms such as mild leaf wilting, due to the level of stress imposed (25% of pot capacity)
- In the heat map (Figure 3C) of roots, the expression pattern of D1 seems to be very different than D2 and D3, but the PCA plot does not show much variation. Authors should explain this. Is D1 an outlier?
Reply: Although D1 is different from D2 and D3, D1 is less different from D2 and D3 than to the other treatments, so they are closer in the PCA analysis. In other words, the differences among the replicates are smaller than the differences among the treatments, which allowed statistical validation.
- Many proteins were found to be downregulated with both single and combined stress. This reviewer thinks that Authors did not focus much on the downregulated proteins.
Reply: Most of the proteins decreased in our work are involved in common plant metabolic processes that have already been widely described in the literature as catalytic activity, hydrolase activity, binding proteins, among others. In addition, a large amount of proteins involved in photosynthesis were reduced in our study and considering the importance of this process for plant survival and yield, they were selected and discussed in the text. We focused on increased proteins mainly because of their relevance in biological processes related to nematode parasitism and lack of water.
- ‘Differentially abundant’ seems to be misleading. Instead, ‘differentially regulated’ could be used. Also, the proteins that are increased in expression when compared to control can be mentioned as upregulated and proteins that are decreased in expression can be mentioned as downregulated.
Reply: In fact, in proteomics, the terms “regulated” and “expressed” should be avoided, since only genes are regulated/expressed and proteins are synthesized/produced. This is often a recommendation in proteomics publications and therefore we have been using the term “differential abundance”.
- What cutoff conditions were set to choose the differentially regulated proteins? For example, p value, fold change etc. Please mention it in the manuscript.
Reply: Proteins were considered differentially abundant when presenting ANOVA p < 0.05 and fold change ≥1.1. This information was added in line 214, new version.
- It is surprising that protein in response to stimulus is unaltered in nematode group. Is there a possible explanation for this?
Reply: In general, it is not possible to state that there are no proteins altered in the biological process of response to stimuli, but perhaps they just have not been identified as differentially abundant according to our experimental design.
- Authors could use cytoscape analysis for visualizing the PPI network and network analysis of differentially expressed genes.
Reply: Cytoscape analysis would be ideal for visualizing and analyzing networks, however, time is short to analyze data and write key results.
- Authors can explain in detail how the knowledge gained from this study can be used in dealing with the biotic and abiotic stress confronted by other plants.
Reply: The increasingly frequent occurrence of climate changes requires new strategies, such as the search for new genes to be used in plant genetic engineering. Through the proteins discovered in this study, it is possible to identify their coding genes and transform other plants to obtain resistant cultivars. Furthermore, our work can help in the study of other cultivars, affected by the same stresses, which do not have previous molecular studies.

Round 2
Reviewer 2 Report
Dear Authors,
Thanks for considering the suggestions and modifying the manuscript